# Biomarkers from Peri-Implant Crevicular Fluid (PICF) as Predictors of Peri-Implant Bone Loss: A Systematic Review

**DOI:** 10.3390/ijms24043202

**Published:** 2023-02-06

**Authors:** Francesca Delucchi, Camilla Canepa, Luigi Canullo, Paolo Pesce, Gaetano Isola, Maria Menini

**Affiliations:** 1Department of Surgical Sciences (DISC), Division of Prosthodontics and Implant Prosthodontics, University of Genoa, Largo R. Benzi 10, 16132 Genova, Italy; 2Unit of Periodontology, Department of General Surgery and Surgical-Medical Specialties, University of Catania, 95124 Catania, Italy

**Keywords:** dental implants, bone loss, biomarkers, peri-implant crevicular fluid, systematic review

## Abstract

The aim of the present systematic review is to summarize current knowledge regarding the analysis of biomarkers extracted from peri-implant crevicular fluid (PICF) as predictors of peri-implant bone loss (BL). An electronic search was conducted on three databases, PubMed/MEDLINE, Cochrane Library, and Google Scholar, to find clinical trials published until 1 December 2022 suitable to answer the following focused question: in patients with dental implants, are biomarkers harvested from PICF predictive of peri-implant BL? The initial search yielded a total of 158 entries. After a full-text review and application of the eligibility criteria, the final selection consisted of nine articles. The risk of bias in included studies was assessed using the Joanna Briggs Institute Critical Appraisal tools (JBI). According to the present systematic review, some inflammatory biomarkers harvested from PICF (collagenase-2, collagenase-3, ALP, EA, gelatinase b, NTx, procalcitonin, IL-1β, and several miRNAs) seem to be correlated with peri-implant BL and may assist in the early diagnosis of pathological BL, that characterizes peri-implantitis. MiRNA expression demonstrated a predictive potential of peri-implant BL that could be useful for host-targeted preventive and therapeutic purposes. PICF sampling may represent a promising, noninvasive, and repeatable form of liquid biopsy in implant dentistry.

## 1. Introduction

Fixed implant-supported restorations are extensively employed to rehabilitate partially or completely edentulous patients with predictable outcomes. However, despite their high percentages of success, dental implants are not free from possible complications with consequent failure, the causes of which are still the object of debate in the dental scientific community.

In particular, peri-implant infections are multifactorial pathological conditions characterized by peri-implant mucosal inflammation with or without progressive loss of supporting bone (i.e., peri-implantitis or peri-implant mucositis, respectively) [1]. Peri-implantitis may be asymptomatic or may appear clinically as mucosal erythema, edema, increased probing depth (PD), bleeding on probing (BOP) with eventual suppuration, and nonlinear progressive bone loss (BL) [2].

The diagnosis of peri-implantitis, especially in its early phases, is crucial in order to prevent the need to treat an active pathology, as an effective and predictable treatment protocol has not been universally validated [3]. In addition, diagnosis of peri-implantitis is not easy, and different criteria have been proposed by different authors [4,5]. According to the last ITI Consensus Report, the presence of BOP is not always predictable for the presence of peri-implantitis, and BOP alone is insufficient for making a diagnosis [6]. 

Furthermore, probing an implant can be useful to monitor PD, but it may be insufficient to determine the extent and pattern of BL over time without radiographs [3].

Indeed, the most frequently used definition of peri-implantitis considers it an “inflammatory reaction associated with loss of supporting bone tissue around an implant” [7]. Accordingly, the new definition of peri-implantitis proposed by Renvert et al. was based on the concomitant presence of peri-implant signs of inflammation and radiographic BL following initial healing [8]. 

However, radiographical peri-implant bone level assessment is not always predictable and presents several limitations, including that only mesial and distal BL can be evaluated in periapical and panoramic radiographs. Dedicated software can be employed to measure bone level change, as well as implant length can be used to correct the radiographic distortion. However, it is possible that not all lesions will be identified, leading to a lack of sensitivity [9]. 

Moreover, even if all clinical parameters and changes in bone levels were combined, they may not be sufficient to predict the patient’s risk of developing peri-implantitis and its prognosis at the beginning of the inflammation process [10]. For this reason, early diagnosis of pathologic BL and identification of early biomarkers for a peri-implant disease are essential. Diagnosis could be implemented by detecting immunological host-derived molecules, such as chemokines, cytokines, bone markers, and enzymes involved in peri-implant tissues turnover [11]. Biomarkers such as pro-inflammatory cytokines (i.e., tumor necrosis factor-alpha (TNF-α), interleukin-1 beta (IL-1β), interleukin-6 (IL-6), and interleukin-17 (IL-17)) are classically associated with the initiation of the inflammatory cascade. They have also been proven to be stimulated by periodontal pathogens’ virulence. Other substances, including neutrophil elastase collagenase, alkaline phosphatase, and aspartate aminotransferase, have been weakly associated with peri-implantitis [12]. Markers of bone tissues (i.e., osteoprotegerin (OPG) and soluble receptor activator of nuclear factor kappaB Ligand (sRANKL)), osteoclastogenic-related cytokines and chemokines (i.e., granulocyte colony-stimulating factor (G-CSF), matrix metalloproteinase-8 (MM-8), monocyte chemoattractant protein (MCP-1) are other important molecules which could be considered to understand better the immune-inflammatory profile of peri-implant disease [13].

More recently, thanks to the development of genomics and epigenomics, other classes of biomarkers, which may help identify individual susceptibility, have been considered for studying multifactorial and complex diseases. MicroRNAs (miRNAs), for example, are small endogenous sequences of noncoding RNAs (ncRNAs) responsible for specific regulation of gene expression in a post-transcriptional manner [14]. They are involved in biological processes, such as immune-inflammatory response, bone metabolism, cell replication, and apoptosis [15]. They are already extensively employed for the early diagnosis, prognosis, and personalized therapies of oncologic and genetic diseases, but they have still been scarcely explored in dental implantology [16].

Interestingly, specific expression profiles of miRNAs extracted from peri-implant tissues have been reported to be predictive of specific clinical outcomes of dental implants and may be used as biomarkers in implant dentistry with diagnostic and prognostic purposes [14,17,18]. Although a mini-invasive sample of peri-implant tissue might be sufficient, this procedure shows a certain degree of invasiveness.

It is thought that the detection of biomarkers in several biologic fluids may be a predictable surrogate of traditional tissue biopsies for diagnosis and prognosis of inflammatory processes, and it has been demonstrated that peri-implant disease could be effectively assessed by the analysis of peri-implant crevicular fluid (PICF) from the peri-implant pocket [19,20]. 

In the last decades, several studies have shown the presence of host-derived biochemical mediators in PICF, and levels of these inflammatory molecules have been proposed as a measure of active peri-implantitis. Biomarkers assessment in PICF may also be useful to identify specific markers responsible for the onset and development of peri-implantitis since its earliest stages when it is still clinically latent.

As a further advantage, PICF is also a site-specific and easily collectible biofluid that could be valuable for the examination of immunological biomarkers by a noninvasive method, that might be repeated over time, besides the fact that peri-implantitis is usually accompanied by an increased volume of PICF [21].

A recently published study demonstrated that many of the miRNAs extracted from PICF were common to those detected in soft tissue taken from the same peri-implant sites, supporting the hypothesis that PICF is a valuable substitute for peri-implant soft tissue samples [20]. 

However, the scientific literature is not conclusive on the matter. Some systematic reviews on the most common cytokines released in PICF in healthy and peri-implantitis sites are present, but they concluded that the scientific evidence is limited on the topic [22,23,24]. In addition, none of them is focused on the correlation between peri-implant BL (the clinical parameter characterizing peri-implantitis) and biomarkers taken from PICF, besides the fact that none include miRNAs among the biomarkers investigated.

In order to shed light on the topic and contribute to the development of clinical indications for the diagnosis and prognosis of peri-implant disease, the present systematic review aims to summarize current knowledge regarding the analysis of biomarkers extracted from PICF as predictors of peri-implant BL.

## 2. Methods 

The present systematic review adhered to the Preferred Reporting Items for Systematic Review and Meta-Analyses (PRISMA) Statement [25] and was registered in PROSPERO, the international prospective register of systematic reviews (Prospero ID: CRD42022381691).

The aim of the review was to shed light on the possible correlation between biomarkers detected in peri-implant crevicular fluid (PICF) and peri-implant BL. Therefore, the following focused question was established according to PI(C)O strategy: In patients with dental implants (P), are biomarkers harvested from crevicular fluid (I) predictive of peri-implant BL (O)?

-Population: patients with dental implants;-Intervention: sampling biomarkers from PICF in patients with peri-implant BL;-Outcomes: correlation between biomarkers in PICF and peri-implant BL.

### 2.1. Search Strategy

The search strategy included the examination of electronic databases, supplemented by hand searches. The National Library of Medicine database (MEDLINE) through its online search engine (PubMed), Cochrane Library, and Google Scholar were the internet sources explored through advanced searches in order to find papers that satisfied the study purpose.

The last search was performed on 1 December 2022.

The following word combination was employed and adapted for each database: “dental implants” [MeSH], “peri-implant crevicular fluid” or PICF, and “biomarker” or mi-RNA.

Possible additional search items were searched by changing the word mi-RNA to miRNA or micro-RNA, or microRNA or cytokine, and combining the firstly typed keywords with “peri-implantitis” or “peri-implant disease” or “bone loss” or “bone resorption”. 

Finally, a hand search was performed by screening the reference list of all included publications to select potentially relevant additional studies. 

All the original studies investigating a correlation between peri-implant BL and biomarkers collected from PICF were included if they met the following inclusion criteria: Studies conducted on humans;Studies that included radiographic measurement of peri-implant bone resorption;Studies with quantification of any type of biomarker extracted from PICF.

Eligible articles included: comparative studies, cohort studies, case-control studies, clinical trials, cross-sectional, retrospective and prospective studies, and case series. Only studies with available full text were taken into consideration. Any full text not available on the internet databases was asked to the corresponding author by e-mail.

In vitro studies, animal studies, as well as nonoriginal studies (i.e., narrative or systematic reviews, letters, editorials, expert opinions, etc.), and redundant publications were excluded. Finally, the reference list of pertinent systematic reviews was also checked in order to look for possible additional studies to be included.

No restrictions in the year of publication were applied, while restrictions in terms of language were used: only papers written in Italian or English were included. 

Other exclusion criteria were as follows:Studies that evaluated biomarkers in tissue, serum, saliva, and other biological sources but not in PICF;Impossibility of correlating the investigated biomarkers with peri-implant BL;Fluid collection during early osseointegration: a minimum of 3 months post-implant insertion was necessary.

Titles and abstracts of the selected papers were screened by two independent reviewers (F.D. and C.C.) for possible inclusion. The full texts of all studies of possible relevance were then obtained for independent assessment by the reviewers. Disagreements between reviewers were resolved by discussion between the two review authors; if no agreement could be reached, a third author decided (M.M.).

The results of the study search and selection are reported in the flow diagram in Figure 1.

### 2.2. Data Extraction

The following data were extracted and collected in an Excel sheet: author(s), publication year, journal, study design, number of patients, number of implants, number of samples, bone loss, assessed biomarkers taken from PICF, methodology used for the analysis, and main outcomes.

### 2.3. Quality Assessment 

Two reviewers (F.D. and C.C.) have been independently involved in the quality assessment of the selected studies using the Joanna Briggs Institute Critical Appraisal tools (JBI) [26]. In particular, two different checklists were used for cross-sectional and cohort studies, respectively.

JBI checklist for cohort studies included the following questions:Were the two groups similar and recruited from the same population?Were the exposures measured similarly to assign people to both exposed and unexposed groups?Was the exposure measured in a valid and reliable way?Were confounding factors identified?Were strategies to deal with confounding factors stated?Were the groups/participants free of the outcome at the start of the study (or at the moment of exposure)?Were the outcomes measured in a valid and reliable way?Was the follow-up time reported and sufficient to be long enough for outcomes to occur?Was follow-up complete, and if not, were the reasons for the loss of follow-up described and explored?Were strategies to address incomplete follow-up utilized?Was appropriate statistical analysis used?

JBI checklist for cohort studies included the following questions:Were the criteria for inclusion in the sample clearly defined?Were the study subjects and the setting described in detail?Was the exposure measured in a valid and reliable way?Were objective, standard criteria used for measurement of the condition?Were confounding factors identified?Were strategies to deal with confounding factors stated?Were the outcomes measured in a valid and reliable way?Was appropriate statistical analysis used?

Based on the potential risk of bias, the following answers were possible for each of the mentioned domains: “Yes”, “No”, “Unclear”, or “Not applicable”.

## 3. Results and Discussion

### 3.1. Bibliographic Search and Study Selection 

The initial search strategy provided a total of 158 articles: 101, 17, and 40 articles were found on PubMed/MEDLINE, Cochrane Library, and Google Scholar databases, respectively. After eliminating all duplicates, 127 possibly relevant studies were detected. In fact, a total of 31 duplicate articles were removed before the screening. After screening the articles’ titles and abstracts, 38 possibly relevant studies were detected for full-text examination. The final selection after full-text analysis included nine papers [12,14,27,28,29,30,31,32,33]. A flowchart was drawn up describing the results of the study search and selection (Figure 1).

### 3.2. Description of Included Studies 

The nine included studies have been conducted from 2000 to 2022. One of the included studies was conducted in Turkey [29], two in Finland [27,28], one in Sweden [32], one in Iran [30], one in Saudi Arabia [33], one in Japan [31], one in Switzerland [12], and one [14] in Italy. All the studies were written in English. 

Detailed information for the nine included studies is reported in Table 1.

Participants in all the included studies were in good general health and had not received any medication that could influence the peri-implant pathological process. All the studies specified the number of implants investigated, and some also specified the division into groups: healthy, affected by mucositis, or peri-implantitis. Healthy implants were defined as such when they showed no bone resorption and no signs of inflammation. However, the definition of pathological peri-implant BL was different between the articles. Five studies categorized “pathologic” BL in mm: BL ≥ 3 mm [33], BL > 3 mm [27,28], and BL > 2.5 mm [31]. Menini et al. considered “augmented” (versus “normal”) a BL > 1 mm [14]. In one study, peri-implantitis was considered present if crestal BL was greater than 20% of the implant length in at least one site (mesial or distal) along the implant [12]. Finally, three studies did not specify any millimetric cut-off value to define an augmented or pathologic BL [29,30,33].

BL measurements were calculated from intraoral radiographs in all the included investigations. 

Eight of the included studies were cross-sectional studies [12,27,28,29,30,31,32,33]. Only the study by Menini et al. was a prospective cohort study [14].

The following biomarkers were examined: calprotectin, cross-linked N-telopeptides of type I collagen (NTx), cathepsin-K, alkaline phosphatase activity (ALP), elastase activity (EA), inhibitor α2-macroglobulin (α2M), procalcitonin, interleukins (IL-1β and IL-34), monocyte chemotactic protein-1 (MCP-1), collagenase-2, collagenase-3, gelatinase b, colony-stimulating factor (CSF-1), and miRNAs. MiRNAs were evaluated in one unique study using a microarray testing the expression of all 2700 human miRNAs [14]. In this cohort study, miRNAs were extracted from PICF, analyzed, and they were considered predictive of BL. 

Plagnat et al. evidenced the correlation between some specific biomarkers in PICF (elastase, α2-macroglobulin, and alkaline phosphatase) and BL around implants [12].

Lira et al. considered the correlation between three other biomarkers (CSF-1, IL-34, IL-1β) and BL [32].

The study by Yaghobee et al. assessed the correlation between IL-1β level and BL [30], while in the article of Sakamoto et al., the association between calprotectin and cross-linked N-telopeptides of type I collagen and BL rate I was compared in sites affected by peri-implant disease and healthy sites [30].

In contrast, in the comparative study of Yamalik et al., no correlation was found between the enzymatic profile of PICF (i.e., cathepsin-K) and BL measurements [29]. In the study of Algohar et al., procalcitonin was collected from different sites to find a link with BL [33].

Then, there are two studies Ma et al. [27,28] on the same sample of patients. They were both included since they reported the outcomes regarding different biomarkers: in the first study, the difference between activated and total gelatinase B levels was measured in three different BL groups (Group 1: BL < 1 mm; Group 2 BL from 1 to 3 mm; Group 3 BL > 3 mm), while in the other paper collagenase-2 and collagenase-3 levels were evaluated in the same BL groups. 

In five articles, PICF collection was obtained using paper points left in the sulcus for 30 s [14,29,30,31,32]. In two studies, a filter strip was placed into the sulcus for 4 min [27,28]. In one study, PICF was collected with paper strips left in the peri-implant sulcus for 15 s [12]. Finally, in one study, paper cones were employed, but the insertion time was not specified [33].

In most of the studies, PICF samples were collected at mesial and distal sites. In the study by Lira et al., it was specified that the sites were inflamed with gingival indexes of 1 or 2 [32]. In the studies by Ma et al., the samples were collected from the site with maximum vertical BL and assessed by X-ray [27,28].

Biomarkers collected from PICF were examined through different types of assays: enzyme-linked immunosorbent assay (ELISA) [12,30,31,33], cathepsin-K activity assay kit [29], P-nitrophenyl-phosphate as substrate [12], low molecular weight fluorogenic substrate [12], modified urokinase assay [28], time-resolved immunofluorometric assay, quantitative immunoblot [27], and commercial enzyme-linked immunosorbent assays [32]. MiRNAs were evaluated with microarray technology [14].

Because of the methodological heterogeneity of the included studies, and in particular, because different biomarkers were investigated in the different studies, a meta-analysis was not appropriate and was not conducted.

### 3.3. Results of Individual Studies

Most of the studies found a positive correlation between the biomarkers collected from PICF and BL. In the study by Sakamoto et al. [31], the BL rate in healthy sites ranged between 6.9 and 41.8%, while that in diseased sites was between 7.7 and 80.0%; a positive correlation was observed between NTx amounts and BL rate (ρ = 0.570, *p* < 0.001). In the study by Algohar et al. [33], implants were divided into healthy and diseased (with mucositis or peri-implantitis), and it was found that mean BL in healthy, mucositis, and peri-implantitis groups were, respectively: 0.7 mm (0.5), 1.1 mm (0.6), and 2.5 mm (0.9). In the peri-implantitis group, a significant positive correlation was observed between crestal BL (*p* = 0.0013) and PICF procalcitonin levels. 

In the study by Menini et al. [14], there was a positive correlation between miRNA expression profile and BL. Moreover, 14 miRNAs that were altered in PICF in the case of greater BL were also altered in the soft peri-implant tissue of the same implant sites. 

In the study by Ma et al. [27], it was found that collagenase-2 (*p* = 0.049) and collagenase-3 (*p* = 0.041) levels were significantly higher in Group 3 than in Group 1 and 2 (Group 1: BL < 1 mm; Group 2 BL from 1 to 3 mm; Group 3 BL > 3 mm). The authors concluded that collagenase-3 and collagenase-2 produced by adjacent bone osteoclast cells reflect irreversible peri-implant vertical BL around loosening dental implants. In the second study by Ma et al. [28], the differences between activated (*p* = 0.044) and total gelatinase B (*p* = 0.026) levels were significant in the three BL groups. Furthermore, gelatinase B levels were increased in Group 3 compared to Groups 2 and 3. Activation of gelatinase B together with elevated mGI (modified Gingival Index) eventually reflects active phases of peri-implantitis and may prove to be diagnostically useful. 

In the study of Plagnat et al. [12], implants were divided into two groups: healthy implants that lost a mean maximum of 0.6 mm of crestal bone, whereas in the diseased group (that is, implants with a BL greater than 20% in at least one site-mesial or distal-along the implant), a correlation was found between EA and the percentage of BL. 

Additionally, in the study by Algohar et al. [33], in the peri-implantitis group, a significant positive correlation was observed between crestal BL (*p* = 0.0013) and PICF procalcitonin levels. In fact, a significantly higher level of PICF procalcitonin was found in the implants with BL compared to the groups of healthy implants (*p* = 0.039) and groups of implants with mucositis (*p* = 0.042). Even the study by Yaghobee et al. found a positive correlation between IL-1β levels and BL (*p* < 0.0001) [30].

Only two studies did not find any statistically significant correlation between BL and PICF biomarkers. In Lira J et al., CSF-1, IL-34, and IL-1β were not related to BL [33]. Moreover, in the article by Yamalik et al., the mean BL values for subgroups P-1, P-2, and P-3 were 1.242 mm, 1.514 mm, and 1.844 mm, respectively (*p* = 0.087), and mean total cathepsin-K activity levels were 3.637, 6.114, and 16.290 units, respectively, with no positive correlation between the enzymatic profile of PICF and BL [29].

**Table 1 ijms-24-03202-t001:** Main characteristics of included studies.

Authors	Study Design	Number of Patients	Number of Implants	Assessed PICF Biomarkers	Definition of BL *	Type of Assay	Main Outcomes
Ma et al., 2000 [27]	Cross- sectional study	13	49	Collagenase-2 Collagenase-3	Group 1: BL < 1 mm; Group 2: BL from 1 to 3 mm; Group 3: BL > 3 mm	Time-resolved immunofluorometric assay (collagenase-2) Quantitative immunoblot (collagenase-3)	Collagenase-2 (*p* = 0.049) and collagenase-3 (*p* = 0.041) levels were significantly higher in Group 3 than in Groups 1 and 2. Collagenase-3 and collagenase-2 produced by adjacent bone osteoclast cells reflect irreversible peri-implant vertical BL around loosening dental implants. Measurements of collagenase-3 and collagenase-2 could be used as markers to indicate the degree of peri-implant vertical BL.
Plagnat et al., 2002 [12]	Cross- sectional study	15	19 (healthy: 11; peri-implantitis: 8)	ALP EA α2M	Healthy implants: no radiographic evidence of BL. Implants with peri-implantitis: crestal BL greater than 20% in at least one site (mesial or distal) along the implant	P-nitrophenyl-phosphate as substrate (ALP) low molecular weight Fluorogenic substrate (EA) ELISA (α2M)	ALP and EA were correlated with the percentage of BL. ALP and EA could be promising markers of BL around dental implants.
Ma et al., 2003 [28]	Cross- sectional study	13	46	Gelatinase b	Group 1: BL < 1 mm; Group 2: BL from 1 to 3 mm; Group 3: BL > 3 mm.	Modified urokinase assay	The differences between activated (*p* = 0.044) and total gelatinase B (*p* = 0.026) levels were significant in the three BL groups. Furthermore, gelatinase B levels were increased in Group 3 compared to Groups 2 and 3. Activation of gelatinase B together with elevated mGI eventually reflects active phases of peri-implantitis and may prove to be diagnostically useful.
Yamalik et al., 2012 [29]	Cross- sectional study	40	54 (Group P-1: 19 healthy implants; Group P-2: 27 implants with mucositis; Group P-3: 8 implants with peri-implantitis)	Cathepsin-K	The actual distance between two consecutive threads of the dental implant was used as a reference point	Cathepsin-K activity assay kit	Mean BL values for subgroups P-1, P-2, and P-3 were 1.242, 1.514, and 1.844 mm, respectively (*p* = 0.087). Mean total cathepsin-K activity levels of subgroups P-1, P-2, and P-3 were 3.637, 6.114, and 16.290 units, respectively. However, there is no positive correlation between the enzymatic profile of PICF and the BL measurements. Mean BL around dental implants did not significantly correlate with total cathepsin-K activity.
Yaghobee et al., 2013 [30]	Cross- sectional study	32	41	IL-1β	BL measured by intraoral periapical radiographs	Enzyme-linked immunosorbent assay (ELISA)	It seems that there Is a positive correlation between IL-1β level and BL (*p* < 0.0001) Mean BL: 1.66 mm.
Sakamoto et al., 2018 [31]	Cross- sectional study	35	74 (healthy: 34; peri-implantitis: 40)	Calprotectin and cross-linked N-telopeptide of Type I Collagen (NTx)	BL of more than 2.5 or 3 mm evaluated around dental implants by intra-oral radiographs	Enzyme-linked immunosorbent assay (ELISA)	The mean BL rate of peri-implant disease sites was 42.7%, and that of healthy sites was 19.7%. The BL rate in healthy sites ranged between 6.9 and 41.8%, and that in diseased sites was between 7.7 and 80.0%. A positive correlation was observed between NTx amounts and the BL rate (ρ = 0.570, *p* < 0.001). Calprotectin and NTx in PICF are markers of inflammation and BL in peri-implant tissues and may be useful diagnostic markers for peri-implant diseases.
Lira-Junior et al., 2019 [32]	Cross- sectional study	43	42 (mucositis: 20; peri-implantitis: 22)	CSF-1; IL-34; IL-1β	BL measured by intraoral periapical radiographs; mucositis: BL around the implant not reaching the first thread; peri-implantitis: BL involving at least two implant threads	Commercial enzyme-linked immunosorbent assays	There is no statistically significant correlation between CSF-1, IL-34, IL-1β, and BL.
Algohar et al., 2020 [33]	Cross- sectional study	60	94 (healthy: 32; peri-implant mucositis; 27: peri-implantitis: 35	Procalcitonin	BL: radiographic level of bone ≥3 mm apical of the most coronal portion of the intraosseous part of the implant after initial bone remodeling. BL is defined as the linear distance measured from the implant-abutment junction to the most coronal point of the alveolar crest.	Enzyme-linked immunosorbent assay (ELISA)	Mean BL in healthy, mucositis, and peri-implantitis groups were, respectively: 0.7 mm (0.5), 1.1 mm (0.6), and 2.5 mm (0.9). In the peri-implantitis group, a significant positive correlation was observed between crestal BL (*p* = 0.0013) and PICF procalcitonin levels.
Menini et al., 2021 [20]	Prospective cohort study	7	14	MicroRNAs *	BL was considered normal if it was ≤1 mm and increased if it was >1 mm	Microarray technology	MiRNAs may be used as biomarkers of peri-implant bone resorption. The following miRNAs were altered in the case of BL both in PICF and in soft peri-implant tissues: miR34 a; miR100; miR106 a; miR126; miR143; miR146 a; miR181; miR200; miR221; miR223; miR375; miR378; miR429; miR1248.

* The expression of all 2700 human miRNAs was evaluated. Sixty-six miRNAs were found to be altered in PICF in the case of peri-implant BL. Of these 66 miRNAs, 14 were also altered in peri-implant tissue of the same implant sites.

### 3.4. Excluded Studies

Out of 38 papers for which the full text was analyzed, 29 were excluded from the systematic review (Appendix A, Table A1) [13,34,35,36,37,38,39,40,41,42,43,44,45,46,47,48,49,50,51,52,53,54,55,56,57,58,59,60,61]. After full-text reading, the studies were excluded for two main reasons: (1) studies that did not measure BL; (2) studies that did not correlate PICF biomarkers with BL.

### 3.5. Quality Assessment

The risk of bias in the included studies was assessed using the Joanna Briggs Institute (JBI) tools. The purpose of this appraisal is to assess the methodological quality of a study and to determine the extent to which a study has addressed the possibility of bias.

Two different JBI critical appraisal checklists were employed based on the types of studies (cross-sectional or cohort studies). The questions for each checklist and the relative risk of bias are reported in Table 2 and in Table 3.

### 3.6. Discussion

The present systematic review aimed to investigate the current scientific evidence on the possible correlation between peri-implant bone loss and biomarkers in PICF.

BL is considered the main clinical sign of peri-implantitis, often accompanied by a set of other signs such as augmented PD, BOP, edema, etc., that may also be present in mucositis on their own. According to the 2017 World Workshop on the Classification of Periodontal and Peri-implant Diseases and Conditions, peri-implantitis is defined as radiographic bone loss ≥3 mm [2]. Otherwise, if no initial radiographs and probing depth values are available, peri-implantitis is defined as radiographic BL ≥ 3 mm and/or PD ≥ 6 mm, together with profuse bleeding.

However, in daily clinical practice, measuring the clinical parameters of peri-implantitis presents several issues, and the diagnosis would occur at a stage of overt disease. For this reason, identification and estimation of specific inflammatory biomarkers predictive of bone resorption in biological fluids, such as PICF, could serve as a noninvasive liquid biopsy [62] for early diagnosis of peri-implantitis when it is still in the subclinical stage or for the identification of patients at higher risk of developing BL. In fact, in peri-implantitis, there is an intense interaction between peri-implant bone and immunoinflammatory cells (such as polymorphonuclear leukocytes and macrophages) with a resultant release of common cytokines, chemokines, and enzymes involved in bone destruction [63].

In the present systematic review, data from 9 original articles have been systematically analyzed for a total of 13 biomarkers harvested from PICF investigated: calprotectin, NTx, ALP, EA, α2M, procalcitonin, IL-1β, MCP-1, collagenase-2, collagenase-3, gelatinase b, CSF-1, and IL-34. In addition, the study by Menini et al. [14] evaluated miRNAs using a microarray testing the expression of all 2700 human miRNAs. A brief description of the investigated biomarkers is reported below.

Calprotectin is a protein that is produced by leukocytes, macrophages, and epithelial cells, and its level increases in various systemic inflammatory diseases.

NTx derives from the degradation of bone type I collagen in osteoclasts and is a specific biomarker of bone resorption [64]. Sakamoto et al. investigated these two biomarkers and observed a positive correlation between NTx amounts in PICF and BL rate [31]. NTx production is due to the action of a protease, CatK, expressed by active resorbing osteoclasts, which hydrolyzes bone matrix proteins. Although CatK could be a good marker for osteoclastic activity, the study by Yamalik et al. did not find any correlation between its levels in PICF and BL measurements [29].

Among the proteolytic enzymes able to degrade extracellular bone matrix, there is a family of 28 enzymes called matrix metalloproteinases (MMPs). Two studies by Ma et al. [27,28] showed that MMP-8, MMP-9, and MMP-13 (also known as collagenase-2, gelatinase B, and collagenase-3) in PICF were associated with more BL in peri-implantitis sites, indicating that they could be three promising biomarkers for peri-implant osteolysis. This is consistent with previous studies that have observed how, for example, MMP-9 (whose role is to degrade collagenase/cathepsin-cleaved bone collagen fragments) may be a prerequisite for osteoclasts’ recruitment to bone resorption sites [65].

Plagnat et al. [12] analyzed the following three other biomarkers: EA, ALP, and α2M. Only PICF levels of ALP (an enzyme involved in bone metabolism, which has been demonstrated to be elevated in sites affected by peri-implantitis) were found to be correlated with pathological BL. Speaking of EA (a serine protease that is able to degrade important proteins in the periodontium), it is worth specifying that its potential clinical value of activity in PICF may be reduced by the presence of endogenous inhibitors such as α2M. However, a study by Gualini et al. observed that peri-implantitis lesions contained greater proportions of B EA-positive cells than mucositis lesions [66].

Two studies included in the present research analyzed PICF levels of IL-1β, a cytokine released by epithelial cells, dendritic cells, connective tissue fibroblasts, neutrophils, and especially macrophages, with pro-inflammatory functions [30,32]. In fact, it regulates collagenase activity and the degradation of the extracellular matrix.

Previous studies have analyzed the correlation between PICF IL-1β levels and peri-implantitis, with discordant findings about its relation with the development of peri-implantitis [22]. In the present review, Lira-Junior et al. [32] found no statistically significant correlation between BL and IL-1β amounts in PICF, while Yaghobee et al. observed only a weak correlation [29]. Lira-junior et al. also found no evidence of a correlation between IL-34 and CSF-1. Both CSF-1 and IL-34 are expressed by gingival fibroblasts, and their expression can be enhanced by IL-1β and tumor necrosis factor-alpha (TNF-α). They have complementary roles, and IL-34 can substitute CSF-1 in RANKL-induced osteoclastogenesis [67].

IL-1β can also induce procalcitonin release from C-cells of the thyroid gland and neuroendocrine cells of the lungs in response to bacterial toxins.

Procalcitonin is a precursor of calcitonin, a hormone responsible for the balance of calcium balance in the body. A significant positive correlation was observed between crestal BL and PICF procalcitonin levels by Algor et al. [33].

As it is evident from the heterogeneity of the cytokines and enzymes analyzed in the different studies included, it was impossible for the authors to conduct a meta-analysis.

Moreover, it must be pointed out that many studies that did not focus on the specific correlation of PICF biomarkers with BL but rather on the generic presence of peri-implantitis had to be excluded from the selection. In fact, the authors decided to focus on the main and clinically more challenging sign of peri-implantitis, that is, BL, avoiding other confounding factors and the bias related to the different definitions of peri-implantitis applied by different authors. This led to the inclusion of nine studies only in the present review. As a consequence, several biomarkers, such as osteoprotegerin (OPG) and RANK ligand (RANKL), although known to be significantly involved in bone metabolism, were not considered in the investigations selected in the present review.

Among the studies included, a distinction must be underlined in the clinical significance of cross-sectional studies where PICF samples were taken at the same time as radiographs and cohort studies. The majority of the investigations herein included were cross-sectional studies, which have an intrinsic predictive limitation. In fact, in cross-sectional studies, PICF biomarker levels were correlated with BL as measured at the same time point of the PICF sample. In contrast, in cohort studies, such as the one by Menini et al. [14], biomarker levels were correlated with BL measured in radiographs taken years after PICF sampling, thus investigating the predictive value of PICF biomarkers in identifying implants at higher risk of developing BL in the subsequent years.

The investigation by Menini et al. [14] was the first and only study harvesting miRNAs from PICF. PICF samples were taken at 3 months post-implant insertion, and miRNA expression was related to BL measured at the 5-year follow-up. This study was the only cohort study included in the present research, and it demonstrates the potential of certain profiles of miRNA expression to predict specific clinical outcomes such as augmented BL. In addition, 14 miRNAs harvested from PICF that were altered in sites with bone resorption at a distance of 5 years were also altered in soft tissue sampled at the same peri-implant sites at the same time point. MiRNA expression, compared to genomic or transcriptomic biomarkers, might have a higher probability of being related to clinical variables, such as peri-implant bone resorption because miRNAs themselves function as the controllers of gene transcription.

A limitation of the present systematic review is the variability of the definitions of augmented or pathological peri-implant BL employed in the included studies. Moreover, most of the studies referred to average BL without specifying whether the correlation was between the BL of the sampling sites and the PICF-extracted markers taken at the same sites.

It should also be considered that BL was correlated to biomarkers whose concentration in PICF might significantly vary over time. Therefore, the clinical significance of a single-moment PICF collection is limited in cross-sectional studies, considering the cyclic progression of peri-implant disease.

In addition, several clinical variables that have not been taken into consideration in the present study, including implant design, implant surface characteristics, prosthesis design, materials, etc., might have affected the outcomes [68,69].

Talking about PICF collection, this review showed how it represents a new model of so-called “liquid biopsy”, which may be applied to early detection, risk assessment, diagnosis, prognosis, and monitoring of peri-implant disease. In fact, it is an absolutely rapid, painless, mini-invasive, and site-specific sampling procedure so that it may be performed repeatedly over time. Among the limitations of this technique, the minimal sample amount (microliters) must be mentioned. Other biologic fluids, such as saliva or blood, can be harvested in higher amounts but lack site-specificity. Other factors that might influence the results include bleeding in the sulcus and saliva from the surrounding. For this reason, during PICF collection, the implant must be carefully isolated, and paper cones/strips presenting blood contamination must be excluded from the analysis.

## 4. Conclusions

In conclusion, according to the present systematic review, some biomarkers harvested from PICF seem to be correlated with peri-implant bone loss and may assist in the early diagnosis of pathological bone resorption that characterizes peri-implantitis. In particular, miRNA gene expression showed to have a predictive potential that could be useful for host-targeted preventive and therapeutic purposes.

PICF sampling may represent a promising noninvasive, site-specific, and repeatable form of “liquid biopsy”.

## Figures and Tables

**Figure 1 ijms-24-03202-f001:**
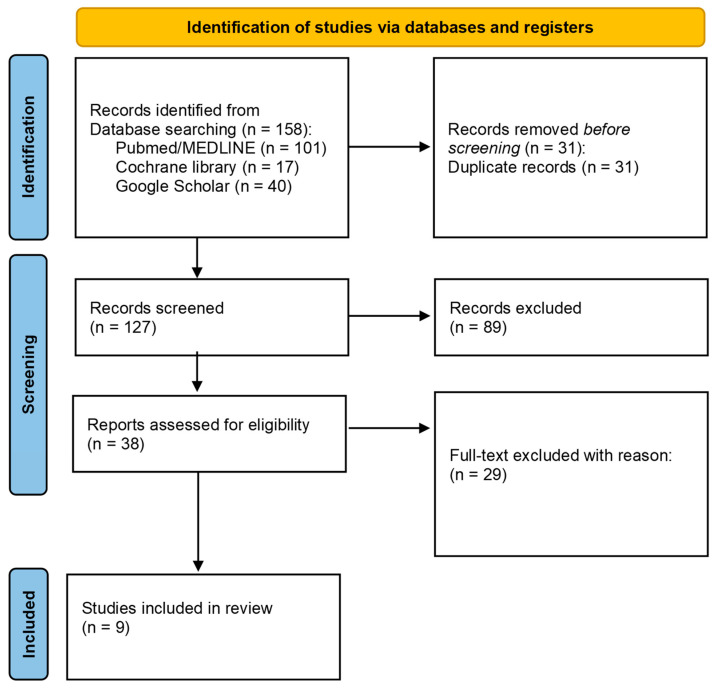
Preferred reporting of systematic reviews and meta-analyses flow diagram related to bibliographic research and study selection.

**Table 2 ijms-24-03202-t002:** Risk of bias for clinical studies included in the present systematic review according to the critical appraisal tools of JBI Scale for analytical cross-sectional studies.

	(1)	(2)	(3)	(4)	(5)	(6)	(7)	(8)
Ma et al., 2000 [27]	No	No	Unclear	Unclear	No	No	Unclear	Unclear
Plagnat et al., 2002 [12]	Yes	Yes	Yes	Yes	Unclear	Unclear	Yes	Yes
Ma et al., 2003 [28]	Yes	Unclear	No	Unclear	Unclear	Unclear	Yes	Yes
Yamalik et al., 2012 [29]	yes	Unclear	Yes	Unclear	Yes	Yes	Yes	Yes
Yaghobee et al., 2013 [30]	Yes	Yes	Unclear	Unclear	Yes	Unclear	Yes	Yes
Sakamoto et al., 2018 [31]	Yes	Yes	Yes	Unclear	Yes	Yes	Yes	Yes
Lira-Junior et al., 2020 [32]	Yes	Yes	Yes	Yes	Yes	Yes	Yes	Yes
Algohar et al., 2020 [33]	Yes	Yes	Yes	Yes	Yes	Yes	Yes	Yes

(1) Were the criteria for inclusion in the sample clearly defined? (2) Were the study subjects and the setting described in detail? (3) Was the exposure measured in a valid and reliable way? (4) Were objective, standard criteria used to measure the condition? (5) Were confounding factors identified? (6) Were strategies to deal with confounding factors stated? (7) Were the outcomes measured in a valid and reliable way? (8) Was appropriate statistical analysis used?

**Table 3 ijms-24-03202-t003:** Risk of bias for clinical studies included in the present systematic review according to the critical appraisal tools of the JBI Scale for Cohort Studies.

	(1)	(2)	(3)	(4)	(5)	(6)	(7)	(8)	(9)	(10)	(11)
Menini et al., 2021 [20]	Unclear	Yes	Yes	Unclear	Unclear	Yes	Yes	Yes	Yes	Yes	Yes

(1) Were the two groups similar and recruited from the same population? (2) Were the exposures measured similarly to assign people to both exposed and unexposed groups? (3) Was the exposure measured in a valid and reliable way? (4) Were confounding factors identified? (5) Were strategies to deal with confounding factors stated? (6) Were the groups/participants free of the outcome at the start of the study (or at the moment of exposure)? (7) Were the outcomes measured in a valid and reliable way? (8) Was the follow-up time reported and sufficient to be long enough for outcomes to occur? (9) Was follow-up complete, and if not, were the reasons for the loss of follow-up described and explored? (10) Were strategies to address incomplete follow-up utilized? (11) Was appropriate statistical analysis used?

## Data Availability

Data available on request.

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
