# Peer review of "Biomarkers from Peri-Implant Crevicular Fluid (PICF) as Predictors of Peri-Implant Bone Loss: A Systematic Review"

_ijms, 2023, doi:10.3390/ijms24043202_

Round 1

Reviewer 1 Report

The present systematic review aimed to investigate the current scientific evidence on the possible correlation between peri-implant bone loss and biomarkers in PICF. It is a very important topic with an increasing number of observational studies that have addressed a potential association between various biomarkers and peri-implant biological complications. The authors use very good and different tools and checklists to quality the studies. Unfortunately, only 9 studies could be included. It seems that this issue should be deeply investigated with high technology tools like the miRNA. The results are well written and clear and also the discussion. I assume that It is important to mention in discussion the limitation of the ELISA method in respect of the PICF collection (limitation of the collection amount, saliva from surrounding, bleeding in the sulcus and etc.) that may influence on the results.

Overall, it is a well written manuscript and very important. I recommend to accept it.

Author Response

Thank you very much for your comments. We have implemented the discussion section as per your suggestion.

"Among the limitations of this technique, the minimal sample amount (microliters) must be mentioned. Other biologic fluids, such as saliva or blood, can be harvested in higher amounts but lack site-specificity. Other factors that might influence the results include bleeding in the sulcus and saliva from surrounding. For this reason, during PICF collection the implant must be carefully isolated and paper cones/strips presenting blood contamination must be excluded from the analysis"

Reviewer 2 Report

Dear Author, congratulations for the article: Biomarkers from peri-implant crevicular fluid (PICF) as predic- 2 tors of peri-implant bone loss: a systematic review.

The article is very good, however, there are many short paragraphs in each section. They should be rewritten in a way showing a greater connection between the phrases, thus making it easier for the reader to understand.

Author Response

Thank you very much for your comments. We have now edited the manuscript creating greater connection between the phrases as per your suggestion.

Round 2

Reviewer 2 Report

All changes have been made.